# Reconfigurable exciton-plasmon interconversion for nanophotonic circuits

Hyun Seok Lee[1,2], Dinh Hoa Luong[1,2], Min Su Kim[1,2], Youngjo Jin[1,2], Hyun Kim[1,2], Seokjoon Yun[1,2] & Young Hee Lee[1,2]

The recent challenges for improving the operation speed of nanoelectronics have motivated research on manipulating light in on-chip integrated circuits. Hybrid plasmonic waveguides with low-dimensional semiconductors, including quantum dots and quantum wells, are a promising platform for realizing sub-diffraction limited optical components. Meanwhile, two-dimensional transition metal dichalcogenides (TMDs) have received broad interest in optoelectronics owing to tightly bound excitons at room temperature, strong light-matter and exciton-plasmon interactions, available top-down wafer-scale integration, and band-gap tunability. Here, we demonstrate principal functionalities for on-chip optical communications via reconfigurable exciton-plasmon interconversions in ∼200-nm-diameter Ag-nanowires overlapping onto TMD transistors. By varying device configurations for each operation purpose, three active components for optical communications are realized: field-effect exciton transistors with a channel length of ∼32 µm, field-effect exciton multiplexers transmitting multiple signals through a single NW and electrical detectors of propagating plasmons with a high On/Off ratio of ∼190. Our results illustrate the unique merits of two-dimensional semiconductors for constructing reconfigurable device architectures in integrated nanophotonic circuits.

[1] Center for Integrated Nanostructure Physics (CINAP), Institute for Basic Science (IBS), Suwon 440-746, Korea. [2] Department of Energy Science, Sungkyunkwan University, Suwon 440-746, Korea. Correspondence and requests for materials should be addressed to H.S.L. (email: hs.lee@skku.edu) or to Y.H.L. (leeyoung@skku.edu).

Photonics for light manipulation have the potential to meet the recent information technology demands of high-speed and massive data processing[1,2]. Electrical switches, wavelength converters and multiplexers are crucial components for integrating optical circuits[2–6]. Quantum well (QW)-based excitonic transistors, in which exciton flux excited by photons diffuses along the QW channel and switched by the gate modulation of channel potential, are promising devices to realize high-speed interconnections[7,8]. Nevertheless, short-lived excitons and the finite exciton binding energy critically limit the operation to low temperatures within the limited channel length of typically less than 3 μm[7,9]. Moreover, a wavelength conversion, that converts an optical signal to a desired wavelength, and a multiplexing technique, that delivers various signals through a single optical guide, are essential to designing the reconfigurable optical communication systems. Although various methods for the wavelength conversion and multiplexing have been developed in conventional technologies based on fibre optics and photonic waveguides, the optical diffraction limit is a fundamental obstacle to reducing the optical component sizes to nanometers[4–6]. Surface plasmon polaritons (SPPs), which are the electromagnetic waves coherently coupled with electron plasma in metals, have the potential to overcome these issues because they allow a strong localization of optical energy at the metal-dielectric interface at a sub-wavelength scale[4,5,10]. Therefore, the use of metallic nanostructures for plasmonic waveguides allows for light manipulation at a nanoscale[1,2,11].

Meanwhile, recently emerged 2D semiconductors have unique merits for nanophotonics[3,10,12,13]. The absence of dangling bonds allows for various combinations of van der Waal heterostructures[14,15] and adaptability with various substrate choices[16], in contrast with the cumbersome epitaxial growth of QWs in limited substrates and processes[7,8,17,18]. The 2D layered structure of transition metal dichalcogenides (TMDs) offers an easy integration for wafer-scale devices[19] and an efficient electrical tunability of optical-electrical properties[3,13], while the top-down approach for arranging quantum dots (QDs) and injecting electrons to individual QDs for optical modulation are still challenging[20–22]. The bandgap is tunable from visible to infrared range by varying the TMDs, alloying[23] and heterostructures[14], which allows a wide-spectral selectivity[10,12]. The device integration and the electrical modulation of exciton fluxes of 2D semiconductors are easier[24] compared to QDs[22]. Tightly bound excitons of TMDs at room temperature allow for room temperature operable excitonic devices[24], which is a stark contrast to QW-based devices operating at a low temperature[7]. Moreover, the efficient excitons-to-plasmon conversion effects with a conversion efficiency as high as ∼32% have been demonstrated in Ag-NW/TMD hybrids[25,26].

Here, we demonstrate the crucial optical components for nanophotonic circuits using reconfigurable exciton-to-plasmon and plasmon-to-exciton interconversions. To realize these concepts, we introduce Ag-nanowires (NWs) for SPP waveguides to 2D semiconductor devices. The Ag-NW waveguides and their

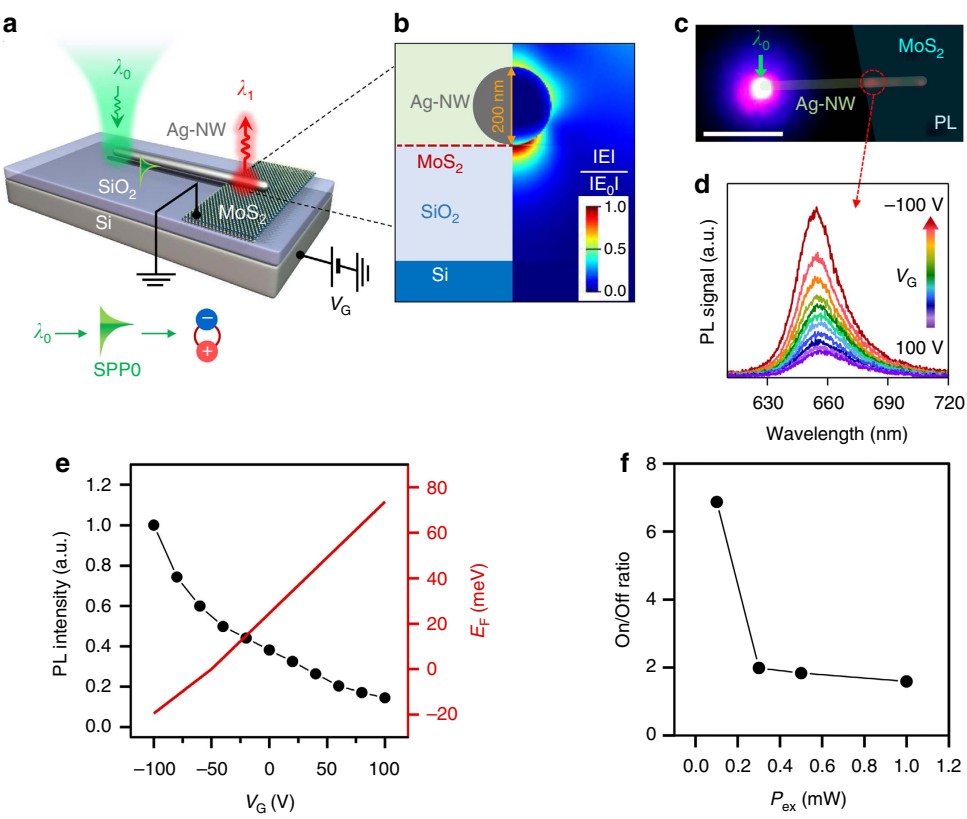

**Figure 1 | Photon-plasmon-exciton conversion and exciton flux modulation. (a)** An Ag-NW overlapped on the monolayer MoS₂ FET. Incident laser ($\lambda_0$) at the end of NW is converted to SPP (SPP0). The propagating SPP0 along the NW is absorbed in MoS₂, and the $\lambda_1$ exciton is generated at the NW/MoS₂ overlapping region. The $\lambda_1$ flux is modulated by $V_G$. **(b)** Cross-section at the NW/MoS₂ overlapping region and optical field map calculated using an FDTD method, implying optical mode confinement near the NW. **(c)** PL image overlaid with false-coloured MoS₂ flakes and the NW. Scale bar: 5 μm. Green arrow: $\lambda_0$ position. Red-dashed circle: PL collection position. **(d)** PL spectra as a function of $V_G$ ranging from −100 V (On state) to 100 V (Off state). **(e)** Integrated PL intensity curve (left axis) and the calculated Fermi level ($E_F$, right axis) as a function of $V_G$. **(f)** On/Off ratio of the integrated intensity as a function of laser power ($P_{ex}$).

hybrids have been well investigated for various purposes because of low ohmic losses and sub-diffraction limited dimensions[2,27,28]. Using Ag-NW-hybridized 2D semiconductor devices with various configurations, three active components for optical communications are realized: field-effect exciton transistors, field-effect exciton multiplexers and electrical detectors of propagating plasmons.

## Results

**Electrical modulation of plasmon-induced exciton flux.** Figure 1a depicts a schematic for the photon-plasmon-exciton conversion process and electrical modulation of the exciton flux. A 200-nm-diameter Ag-NW for SPP waveguides is partially overlapped with a monolayer $MoS_2$ field-effect transistor (FET) on an $SiO_2$ (300 nm)/Si wafer. The laser light ($\lambda_0 = 514$ nm) is focused to the left end of the NW and $\lambda_0$ is coupled to SPP (SPP0) in the NW[20,26] (see Methods). The SPP0 propagates along the NW in the axial direction with a tightly confined optical near-field[20,26,29] (Fig. 1b). The near-field of SPP0 is absorbed in $MoS_2$ layers and excites excitons ($\lambda_1 \approx 660$ nm) at the NW/$MoS_2$ overlapping region[26,30] (see Supplementary Note 1 and Supplementary Fig. 1), where the exciton fluxes are modulated by gate bias ($V_G$) for excess carrier doping[13,24].

Figure 1c shows the photoluminescence (PL) image overlaid with the sample schematic of the $MoS_2$ flake and NW, where the green arrow is the $\lambda_0$ position and the red-dashed circle indicates the PL collection position. Figure 1d shows the PL spectra as a function of $V_G$ ranging from $-100$ V (On state) to 100 V (Off state). The PL intensity increases gradually as $V_G$ is reduced. The Fermi level was calculated from the excess electron and hole densities derived from the back-gate capacitance as a function of $V_G$, as plotted in Fig. 1e, where the charge neutrality point near $-50$ V is taken as a reference[24] (see Supplementary Note 2 and Supplementary Fig. 2), consistent with the intrinsic $n$-type doping state in $MoS_2$. While the Fermi level increases due to the

increased electron carriers at high $+V_G$, the PL intensity gradually decreases in proportion to $E_F$, which is attributed to the Pauli blocking effect for excitons[24]. With increasing $n$-doping ($V_G > -50$ V), the photoexcited electrons are suppressed by Pauli blocking effect and do not contribute to excitonic emission. Consequently, neutral excitons decrease and negative trions increase. Conversely, with increasing $p$-doping ($V_G < -50$ V), the Pauli blocking disappears. As a result, both neutral excitons and positive trions increase, and thus, a total exciton flux for $p$-doping case is larger than that for $n$-doping case (see Supplementary Fig. 3a). Figure 1f shows an On/Off ratio of exciton fluxes as a function of input laser power ($P_{ex}$). The On/Off ratio reaches a factor of 7 at $P_{ex} = 0.1$ mW and details are discussed in Supplementary Note 3 and Supplementary Fig. 3.

**Exciton transistor via exciton-plasmon interconversions.** We demonstrate field-effect exciton transistors with a long channel length ($\sim 32$ μm) via plasmon-to-exciton and exciton-to-plasmon interconversions. Figure 2a depicts the operation principle, where $MoS_2$-FET is located in the middle of the NW. The SPP0 coupled from $\lambda_0$ is absorbed in the $MoS_2$ layer (as discussed in Fig. 1a) and generate exciton ($\lambda_1$). The generated $\lambda_1$ is recoupled to SPP1 and propagates along the NW (Fig. 2a). Finally, the SPP1 is converted to $\lambda_1$ via scattering at the right end of NW (see Methods). Figure 2b shows an optical micrograph and PL image of the hybrid device. The green arrow is the $\lambda_0$ illumination, the red arrow near the NW/$MoS_2$ overlapping region is the $\lambda_1$ emission and the red spot at the right end of the NW (blue-dashed circle) is a PL collection position for Fig. 2c. In the $\lambda_1$ emission region, the propagating SPP0 along the NW is converted to SPP1 via exciton-plasmon interconversion and the propagating SPP1 is monitored by the $\lambda_1$ scattering at the right end of NW. Clear On ($V_G = -100$ V) and Off ($V_G = 100$ V) states of the PL are displayed in Fig. 2c. Notably, the demonstrated channel length reaches $\sim 32$ μm at room temperature, $\sim 10$ times longer that of QW-

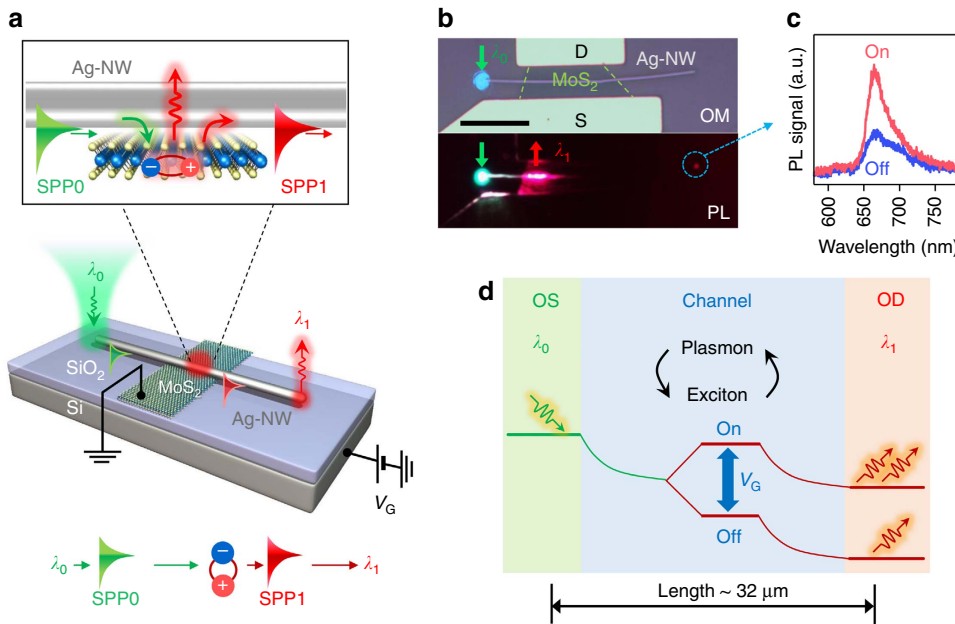

**Figure 2 | Long channel exciton transistor.** (**a**) A long Ag-NW overlapped on the $MoS_2$ FET. The incident $\lambda_0$ is converted to SPP0. The SPP0 propagating along the NW is absorbed in $MoS_2$, and the $\lambda_1$ exciton is generated at the NW/$MoS_2$ overlapping region. The $\lambda_1$ is recoupled to SPP1 near the NW and scattered out at the right end of the NW. The $\lambda_1$ flux is modulated by $V_G$. (**b**) Optical micrograph overlaid with false-coloured $MoS_2$ flakes (top) and PL image (bottom) of the exciton transistor. Source (S) and drain (D) for electrodes of the FET. Green arrow: $\lambda_0$ position. Red arrow: $\lambda_1$ emission. Blue dashed circle: PL collection position. Scale bar: 10 μm. (**c**) PL spectra for On ($-100$ V) and Off (100 V) states. (**d**) Schematic depicts the transistor operation. Optical source (OS): $\lambda_0$ input. Optical drain (OD): $\lambda_1$ output. Channel: NW and NW/$MoS_2$ overlapping region.

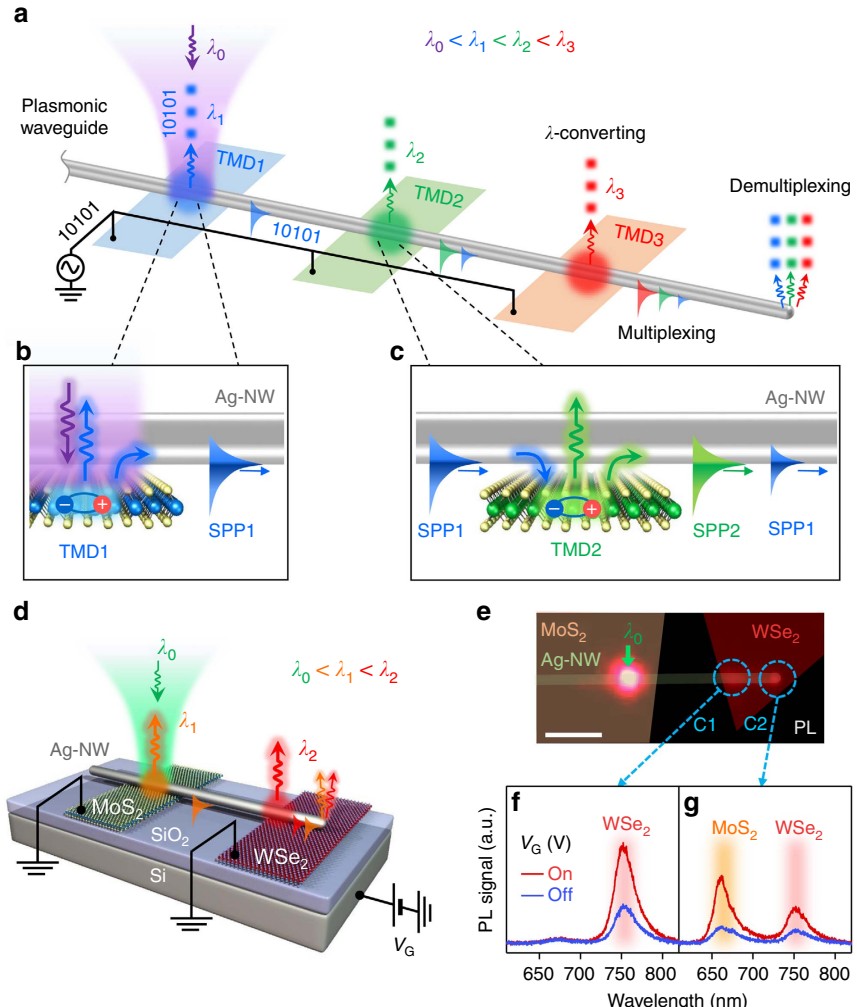

**Figure 3 | Exciton multiplexer.** (**a**) Laterally arrayed TMDs with different bandgaps bridged via an Ag-NW. The exciton flux of each TMD with the corresponding wavelength ($\lambda$) is switched by electrical doping, (**b**) $\lambda_1$ excited by the input light is coupled to SPP1 in the NW, (**c**) propagates along the NW and then excites $\lambda_2$ by partially being absorbed in TMD2 that is coupled to SPP2 in the NW. SPP1 and SPP2 excite $\lambda_3$ in TMD3 and generate SPP3. The three SPP modes deliver the multiplexed wavelengths with optical information generated by electrical modulation. The multiplexed wavelengths are further de-multiplexed by far field scattering at the NW end. (**d**) Schematic of the device consisting of $MoS_2$ and $WSe_2$ monolayers bridged by Ag-NW. Each exciton flux is modulated by $V_G$. (**e**) PL image overlaid with the device structure. The green arrow is the $\lambda_0$ position. C1 and C2 are the PL collection positions. Scale bar: 2 $\mu$m. The PL spectra measured at (**f**) C1 and (**g**) C2 positions for $V_G$ of −100 and 100 V.

based excitonic transistors ($\sim$3 $\mu$m) at low temperatures[7,8]. In the Ag-NW, the corresponding $1/e$ SPP propagation lengths are calculated to 12–31 $\mu$m for 620–760 nm visible light (see Methods), which quantitatively agrees with our experimental results. Even longer propagation length of $\sim$50 $\mu$m at a visible range has been demonstrated in patterned Ag films by the CMOS process on a wafer scale[31]. These length scales are compatible with current nanoelectronics[1]. Moreover, TMDs with smaller bandgaps for long wavelengths can allow even longer SPP propagation. The concept of the field-effect exciton transistor is schematically summarized in Fig. 2d, where the optical source (OS) is the $\lambda_0$ input, the optical drain (OD) is the $\lambda_1$ output and the channel is the NW and NW/$MoS_2$ overlapping region.

**Reconfigurable exciton-plasmon interconversions.** The exciton-plasmon interconversion is more reconfigurable for advanced device architectures. Figure 3a illustrates the operation principle for wavelength multiplexers. Laterally arrayed TMD-FETs, having different bandgaps, are interconnected by an Ag-NW. The excitonic emission wavelength of TMD1, TMD2 and TMD3

corresponds to wavelengths $\lambda_1$, $\lambda_2$ and $\lambda_3$ ($\lambda_1 < \lambda_2 < \lambda_3$), respectively. With the laser ($\lambda_0$) incident on the NW/TMD1 overlapping region, the exciton corresponding to $\lambda_1$ ($> \lambda_0$) is generated from TMD1. The $\lambda_1$ is coupled to SPP1 near the NW (Fig. 3b) without momentum matching via the Förster resonance energy transfer of excitons to plasmons[21,32]. The generated SPP1 propagates along the NW, encounters TMD2 and excites $\lambda_2$, which is partially absorbed in TMD2. The similarly generated $\lambda_2$ propagates along the NW (Fig. 3c). Likewise, the SPP1 and SPP2 excite $\lambda_3$ in TMD3 and generate the SPP3. Eventually, $\lambda_1$, $\lambda_2$ and $\lambda_3$ are multiplexed in the three SPP modes, which are detected at the NW end (see Supplementary Notes 4 and 5 and Supplementary Figs 4 and 5). During this multiplexing process, electrostatic doping of TMD-FETs modulates the exciton fluxes for optical switching information. Notably, the $\lambda_0$ is simply converted to multiple wavelengths via sequential exciton-to-plasmon and plasmon-to-exciton interconversion processes, where excitonic wavelengths are determined by the bandgaps of the TMDs.

Figure 3d shows the device structure and experimental setup required to prove such a concept. Monolayer $MoS_2$ and $WSe_2$

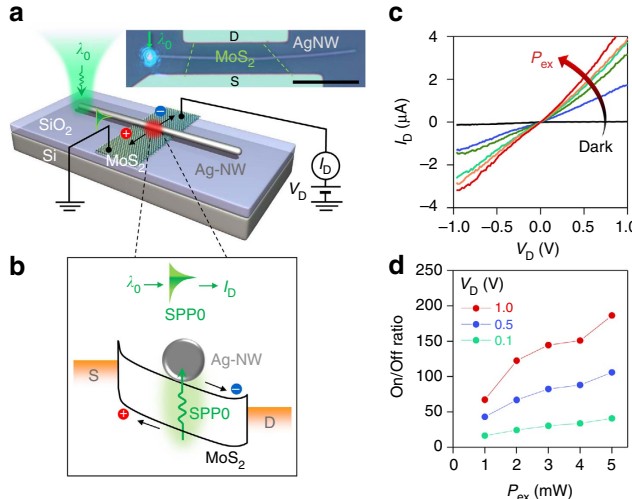

**Figure 4 | Electrical detection of plasmons.** (**a**) An Ag-NW overlapped on the MoS$_2$ FET. Incident $\lambda_0$ at the end of NW is converted to SPP0. The propagating SPP0 along the NW is absorbed in MoS$_2$ and converted to photocurrent ($I_D$) at the NW/MoS$_2$ overlapping region. Scale bar: 10 μm. (**b**) Schematic description of photocurrent generation via plasmon-to-charge conversion at the NW/MoS$_2$ overlapping region. (**c**) $I_D$-$V_D$ curves of MoS$_2$ FET for various $P_{ex}$ ranging from 1 to 5 mW. Black curve: laser-off state (Dark). (**d**) On/Off ratio as a function of $P_{ex}$ for the selected $V_D$ from (**c**).

FETs on SiO$_2$ (300 nm)/Si wafers are interconnected by an Ag-NW. Under the $\lambda_0$ illumination on the NW/MoS$_2$ overlapping region, the generated MoS$_2$ excitons ($\lambda_1 \approx 660$ nm)[24] are coupled to the SPP1. The SPP1 propagates along the NW and excites the WSe$_2$ excitons ($\lambda_2 \approx 760$ nm)[33] at the NW/WSe$_2$ overlapping region (red spot in Fig. 3d) and simultaneously, $\lambda_2$ is recoupled to SPP2. Finally, the SPP1 and SPP2 are multiplexed through the single NW. Figure 3e shows PL image overlaid with the sample image of the TMD flakes and the NW, where C1 and C2 indicate the PL signal collection positions. Figure 3f,g shows the PL spectra that are modulated by $V_G$. The PL spectra for $\lambda_1$ and $\lambda_2$ are clearly observed at C2 (the NW end) via individual plasmon-to-photon conversions from SPP1 and SPP2 to $\lambda_1$ and $\lambda_2$, implying the wavelength de-multiplexing. The $\lambda_1$ scattering at C1 (the NW mid-section) is negligible because the SPP1 is not converted to $\lambda_1$ without a scattering source at the NW mid-section (see Methods). Various combinations of wavelengths are available by rearranging the TMD array and vertical stacking (see Supplementary Notes 4–6 and Supplementary Figs 4–6). In addition, the 2D material dependence on the exciton-plasmon interconversion efficiency was not appreciable, while the strength of the exciton-plasmon interaction strongly relies on the exciton quantum yield of each material (see Supplementary Note 7 and Supplementary Fig. 7). We also investigated the polarization effect of light illumination. A parallel polarization of incident light to an SPP propagation direction along the Ag-NW is essential for the exciton-plasmon interconversions (see Supplementary Note 8 and Supplementary Fig. 8).

**Electrical detection of plasmons with high On/Off ratios.** The propagating plasmons are also detectable by electrical signals, which is another key component for nanophotonic circuits. Figure 4a shows the device structure and experimental setup to prove such a concept using the device for Fig. 2. The SPP0 generated by $\lambda_0$ is absorbed in the MoS$_2$ layer (as discussed in

Figs 1 and 2). The absorbed SPP0 generates electron-hole pairs which are separated to a plasmonic photocurrent via plasmon-to-charge conversion[30,34] under an applied drain bias at the MoS$_2$ FET (Fig. 4b). Figure 4c shows $I_D$-$V_D$ curves of the Ag-NW overlapped MoS$_2$ FET for a dark state and under $\lambda_0$ illuminations at the NW end with various $P_{ex}$. The SPP0 induced photocurrents (for On state) are two orders of magnitude higher than the dark current (Off state). The photocurrent increases with $V_D$ because of an increased drift velocity of photocarriers and a reduced carrier transit time[35]. Figure 4d shows On/Off ratios of $I_D$ in proportion to $P_{ex}$ for various $V_D$. The maximum On/Off ratio reaches $\sim 190$ for $P_{ex} = 5$ mW and $V_D = 1$ V.

## Discussion

In summary, we have demonstrated the crucial optical components for nanophotonic circuits using the reconfigurable exciton-plasmon interconversion and efficient exciton flux modulation of TMDs in various Ag-NW/TMD hybrid architectures at room temperature. The exciton transistor was realized by partially overlapping the Ag-NW on TMD-FETs. The laser-coupled-plasmon propagates through the Ag-NW channel and sequentially excites excitons of TMD-FETs where the exciton flux was modulated by electrical gate doping. Wavelength multiplexing devices were realized by interconnecting Ag-NW to laterally arrayed TMD-FETs having different bandgaps via the sequential exciton-plasmon interconversions. The electrical detections of propagating plasmons along the Ag-NW with a high On/Off ratio were also realized in the Ag-NW hybridized TMD-FET. Our demonstrations of the reconfigurable exciton-plasmon interconversions in various device architectures pave a way to realize various optical components for nanophotonic integrated circuits having advantages of adaptability in wavelength selection and scalability.

## Methods

**Sample preparation.** The MoS$_2$, WS$_2$ and WSe$_2$ monolayer flakes were synthesized on an SiO$_2$ (300 nm)/Si substrate using a chemical vapour deposition method[36]. The synthesized monolayer TMD flakes were transferred onto an SiO$_2$ (300 nm)/Si wafer via the conventional poly(methyl methacrylate) (PMMA) support method[36]. The Ag-NW with a diameter of $\sim 200$ nm (PlasmaChem Corp.; dispersed in isopropyl alcohol) was transferred onto the flake-transferred samples and dried under ambient conditions. The electrical contacts to the TMDs were fabricated by the metal evaporation of Cr/Au (10/50 nm) and e-beam lithography patterning method, where an Si wafer was used to apply the back-gate bias. Finally, the samples were covered with a $\sim 400$-nm-thick PMMA (950 K PMMA, MicroChem Corp., 4% in chlorobenzene) layer to protect the NW from degradation under ambient conditions[25].

**Characterization.** The $I_D - V_G$ measurements in the MoS$_2$-FET were performed at $V_D = 1$ V using an electrical characterization system (Kiethley 4200-SCS, Kiethley Instruments). The PL spectra under the $V_G$ application (Kiethley 6487 picometer/voltage source, Kiethley Instruments) to the Si substrate and the photocurrents generated by plasmons (Agilent B2902A, Agilent Technologies) were measured using a lab-constructed confocal microscope. A laser beam with a wavelength of 514 nm was focused on the sample using an objective lens ($\times 100$, numerical aperture, 0.9). The PL spectra at the desired positions were collected using a pinhole detector under the illumination of the focused laser at other positions, where the laser beam and pinhole positions were controlled by a micromanipulator. The spectra were recorded using a spectrometer and a cooled charge-coupled device camera[25].

**SPP propagation length analysis.** SPP modes for the configuration of PMMA/Ag-NW on an SiO$_2$ (300 nm)/Si substrate were analysed using a finite-difference time-domain numerical simulation (Lumerical Solutions, Inc.) method. The $1/e$ SPP propagation lengths were derived as a function of wavelength from the lowest loss mode among the possible confined modes near the NW[25,26].

**Coupling and decoupling of SPPs at distal ends of Ag-NWs.** To couple the SPP waveguide modes along the Ag-NW axis by the light illumination, the momentum mismatch for longitudinal mode between the SPPs and the incoming photons should be compensated[29,37]. The light scattering can be used to provide additional

wavevectors. When the focused laser light is illuminated at the midsection of the NWs, a cylindrically symmetric shape of the NWs cannot provide the light scattering in the axial direction, and thus, the activation of the SPP waveguide modes along the NW is negligible. However, when light is illuminated at the end of the NW, the light scattering provides additional wavevectors in all directions because the symmetric shape is broken at the distal edge of the NWs. Therefore, the momentum mismatch between the SPPs and the photons is compensated. As a result, the SPP waveguide modes can be activated. In the reverse process, when the propagating SPP modes face the distal end of the NWs, the SPPs are decoupled to radiative emissions but not at the NW midsection[29,37].

**Data availability.** The data that support the findings of this study are available from the corresponding authors H.S.L and Y.H.L on request.

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

## Acknowledgements

This work was supported by the Institute for Basic Science of Korea (IBS-R011-D1).

## Author contributions

H.S.L. designed and developed the work. H.S.L., D.H.L, Y.J., H.K. and S.Y. prepared the samples. H.S.L., D.H.L., M.S.K. and Y.J. performed experiments. H.S.L conducted the numerical simulations. All authors discussed the results. H.S.L. and Y.H.L. analysed and wrote the manuscript.

## Additional information

**Competing financial interests:** The authors declare no competing financial interests.

