## [Peer Review File · Nature Communications]

Reviewers' comments:

Reviewer #1 (Remarks to the Author):

It is an interesting manuscript about the plasmon-exciton interaction. Authors realized plasmon-to exciton, exciton-to plasmon conversion, which demonstrated that plasmon-exciton can be used to the crucial optical components for nanophotonic circuits as interconversions in various NW-hybridized 2D semiconductor devices. The results are interesting and very usefully the applications of plasmon-exciton conversion. It can be accepted after considering below suggestions.

1. Plasmon of 2D layered materials has recently also reported in *Advanced Science*, 2016, 28, 2931.
2. Plasmon-exciton coupling for the ultrasensitive sensor has been reported, see *Light: Science & Applications* (2015) 4, e342.
3. Plasmon waveguide has also been used to the detection of remote SERS and remote plasmon-driven catalytic reaction, see *Light: Science & Applications* (2013) 2, e112; (2014) 3, e199 and *Advanced Science*, 2016, 3, 1500215.

Reviewer #2 (Remarks to the Author):

The paper discusses and shows how exciton-plasmon interactions in Ag/TMD can be integrated into device designs that enables useful interconversions and multiplexing of the signal. The paper discusses several device concepts that layout the basic design strategy for integration of the idea into nanophotonic circuits. The experiments are designed adequately to address and explore a few device properties, predominantly focusing on demonstrating the possibility of signal conversion at different spatially resolved juncture points along and between the Ag SPP waveguide and TMDs. I find this study interesting and connected to the existing research in the area of 2D materials and optoelectronics. However, I would like to raise a few points that I think if addressed can make this paper better and more appropriate for publication in *Nature Communication*.

- 1- The paper does examine about or emphasize on scientifically important aspects of the device properties. Obviously the conversion of the SPP and its interactions with semiconductors in generating excitons is not a new idea. However, the scientifically important aspect perhaps in the material systems examined may be the efficiency of coupling and the effectiveness of modulation. I think the authors can point out why 2D semiconductors may be a more unique choice in development of such device concepts, perhaps pointing at the more effective gate control in these materials for modulation. They also should discuss the efficiency of coupling between SPP and TMDs by examining the yield of the signal generation. This would require few more experiments and if designed well it can make the scientific impact of the paper more significant.
- 2- One fundamental question that I feel needs to be answered is whether the carrier density changes controlled by the gating and its effect on the photoluminescence is the predominant source of changes in the signal or the role it may play in the coupling of the SPP with the TMDs. This can be also understood by better understanding of the yield.
- 3- I think baseline measurements on TMDs and the Ag-NW is very important and is lacking. I suggest that the authors acquire direct PL from the TMDs and examine the SPP propagation characteristics of the Ag-NW independently and report that in the SI.
- 4- Similarly, baseline measurements showing the yield of signal conversion in different Ag-NW TMDs seems useful. It is very useful to show if the strength of the excitonic plasmonic interactions are material system dependent. In addition this will be useful in understanding the role that the non-interacting SPP signal may play in the device architecture and signal propagation in the multiplex systems. I also suggest doing experiments that change the order of the TMDs in the signal propagation. This will be useful in answering some of these questions.

5- Electrical detection device shown in figure 4C is very difficult to understand. I see three electrodes two of them labeled source and drain. The source seems to be connected to the 2 flakes and Drain is connected to one and the third lead is connected to the second flake. Since Ag/NW is a metal the two flakes are also electronically connected. I am not sure what are the potential levels applied on each lead but I think that these measurements are probably done incorrectly and there is a chance that they are simply measuring a photo detection of λ_0 .

At this point I do not recommend the paper for publication but I think with major revisions and addition of new measurements to address some of my concerns this paper can be considered for publication.

Reviewer #3 (Remarks to the Author):

The authors reported the interesting system of a silver nanowire overlapping on the TMDs monolayer field-effect transistor, operating as on-chip optical communication device via exciton-plasmon interconversion. Their experiments seem straightforward and based on their previous results. I wonder if they can provide their answers on my comments and questions below.

1) The authors used the laser excitation and SPP energy as 514 nm, which is quite far from the MoS₂ exciton resonance (660 nm). I wonder if the authors performed the same experiments with the lower laser excitation near MoS₂ exciton, for example, such as a resonant condition between MoS₂ exciton and SPP. If so, I also wonder their results which may help for understanding exciton-SPP interconversion further.

2) I understand that the laser located at the end of the silver nanowire can excite the propagating SPP along the interface between nanowire and dielectrics (SiO₂ or TMDs monolayer) via the scattering of the laser light. However, in case of exciton multiplexer and electrical detection experiments, the author focused the laser at the middle of the silver nanowire, in which there seems no way to compensate the momentum mismatch to excite SPP. If the authors want to claim that SPP can be excited via the emission from the TMDs exciton near silver nanowire, I wonder what kinds of mechanism can explain SPP excitation in this case. Furthermore, it is known that excitons in TMDs are tightly aligned on the plane of the monolayer as described in figure 2a, 3b and 3c, and I guess that the PL from the TMDs excitons will be heading for vertical direction with respect to the monolayer. In this case, the mechanism of SPP excitation is still questionable since it will be similar to the laser excitation shining on the middle of nanowire.

3) The authors explained that the charge neutrality point is -50 V. If this is the case in their experiments, I wonder how to explain the increased PL for the gate bias smaller than -50 V (from -100 V to -50 V) since it is expected the excitonic transition strength being reduced in this condition.

4) If the authors want to emphasize their device mechanism as exciton-plasmon interconversion, polarization (TM, TE modes with respect to the nanowire long axis) dependent experiments are needed to make more clear correlation between exciton and SPP since SPPs in the metal nanowire are known to be sensitive to the TM and TE polarizations.

5) I wonder if the authors have performed any low-temperature experiments other than the reported room temperature results. Two things can be the reasons. At low temperature, exciton dephasing rate is expected to be much lower than the room temperature and the exciton-plasmon interconversion can be enhanced more. Also, by lowering the temperature, the band-gap of TMDs will decrease and the laser exciton and the exciton energy will get closer than the room temperature.

6) There seems to be the summary part missing in the paper.

7) I suggest that the On/Off ratio for the electrical detection experiments should be improved to be published in a high impact journal. I guess that it can be achieved in various ways including reducing the length of the nanowires.

Response to Reviewer #1

It is an interesting manuscript about the plasmon-exciton interaction. Authors realized plasmon-to exciton, exciton-to plasmon conversion, which demonstrated that plasmon-exciton can be used to the crucial optical components for nanophotonic circuits as interconversions in various NW-hybridized 2D semiconductor devices. The results are interesting and very usefully the applications of plasmon-exciton conversion. It can be accepted after considering below suggestions.

Author's answer: We appreciate the reviewer's valuable and constructive comments for our manuscript. The reviewer has already seen our main points and the importance of our manuscript. Thank you very much for your positive comments and concerns. The followings are the details of the responses.

1. Plasmon of 2D layered materials has recently also reported in *Advanced Science*, 2016, 28, 2931.

Author's answer: Unfortunately, we could not find such a paper in the journal of *Advanced Science*. However, we were able to find the exactly same volume number and page in *Advanced Materials*. We believe that it is due to the reviewer's typing error. This reference has reported plasmonic effects of graphene. This is somewhat irrelevant to our work, because the spectral range of graphene plasmon is far-infra red range, while the spectral range of excitons in our study using TMD materials is visible range, which is very different from application point of views.

2. Plasmon-exciton coupling for the ultrasensitive sensor has been reported, see *Light: Science & Applications* (2015) 4, e342.

Author's answer: This reference has reported on the sensor applications using localized surface plasmon assisted surface-catalyzed reactions. Localized-plasmon-driven graphene-assisted sensing effects were monitored by surface plasmon enhanced Raman spectroscopy (SERS) method. However, this reference is also irrelevant to our work, because our study demonstrates optical components for *nanophotonic circuit applications* using exciton-plasmon interconversion effects based on surface plasmon propagation in Ag nanowires and excitons in TMDs.

3. Plasmon waveguide has also been used to the detection of remote SERS and remote plasmon-driven catalytic reaction, see *Light: Science & Applications* (2013) 2, e112; (2014) 3, e199 and *Advanced Science*, 2016, 3, 1500215.

Author's answer: These references have reported remote SERS effects using surface plasmon propagation in Ag nanowires. Although surface plasmon propagation effects of Ag nanowires were used to sense the Raman signal, Raman monitoring is irrelevant to our work. Nevertheless, we introduced these works in introduction part because these works are also important examples for application of plasmon waveguide based on Ag nanowires. To satisfy the reviewer's suggestion, we added the regarding sentence in introduction part (page 3):

“The Ag-NW waveguides and their hybrids have been well investigated for various purposes because of low ohmic losses and sub-diffraction limited dimensions^{2,27,28}”

Moreover, we also cited the regarding reference in Supplementary Note 8:

“These results are consistent with the polarization effects in Ag-NWs in previous reports¹².”

Overall, while some references suggested by reviewer are somewhat different field for citation, we were able to find valuable references, thanks to the reviewer’s suggestion. We have made all necessary revisions based on these comments in introduction part, which made our manuscript stronger.

Response to Reviewer #2

The paper discusses and shows how exciton-plasmon interactions in Ag/TMD can be integrated into device designs that enables useful interconversions and multiplexing of the signal. The paper discusses several device concepts that layout the basic design strategy for integration of the idea into nanophotonic circuits. The experiments are designed adequately to address and explore a few device properties, predominantly focusing on demonstrating the possibility of signal conversion at different spatially resolved juncture points along and between the Ag SPP waveguide and TMDs. I find this study interesting and connected to the existing research in the area of 2D materials and optoelectronics. However, I would like to raise a few points that I think if addressed can make this paper better and more appropriate for publication in Nature Communication.

Author's answer: We appreciate the reviewer's valuable and constructive comments for our manuscript. The reviewer has already seen our main points and the importance of our manuscript. Thank you very much for your positive comments and concerns. The followings are the details of the responses.

1- The paper does examine about or emphasize on scientifically important aspects of the device properties. Obviously the conversion of the SPP and its interactions with semiconductors in generating excitons is not a new idea. However, the scientifically important aspect perhaps in the material systems examined may be the efficiency of coupling and the effectiveness of modulation. I think the authors can point out why 2D semiconductors may be a more unique choice in development of such device concepts, perhaps pointing at the more effective gate control in these materials for modulation.

Author's answer: We already described this point in introduction part.

“Meanwhile, recently emerged 2D semiconductors have unique merits for nanophotonics^{3,11-13}. The absence of dangling bonds allows for various combinations of van der Waal heterostructures^{14,15} and adaptability with various substrate choices¹⁶, in contrast with the cumbersome epitaxial growth of QWs in limited substrates and processes^{7,8,17,18}. The 2D layered structure of TMDs offers an easy integration for wafer-scale devices¹⁹ and an efficient electrical tunability of optical–electrical properties^{3,12}, while the top-down approach for arranging quantum dots (QDs) and electron injection to individual QDs for optical modulation are still challenging²⁰⁻²². The bandgap is tunable from visible to infrared range by varying the TMDs, alloying²³, and heterostructures¹⁴, which allows a wide-spectral selectivity^{11,13}”

Nevertheless, to satisfy the reviewer's suggestion, we emphasized this point by adding the regarding sentence in introduction part (page 2-3):

“The device integration and the electrical modulation of exciton fluxes of 2D semiconductors are easier²⁴ compared to QDs²². Tightly bound excitons of TMDs at room temperature allow for room temperature operable excitonic devices²⁴, which is a stark contrast to QW-based

devices operating at a low temperature⁷.”

They also should discuss the efficiency of coupling between SPP and TMDs by examining the yield of the signal generation. This would require few more experiments and if designed well it can make the scientific impact of the paper more significant.

Author’s answer: As the feasibility study for optical communication applications using 2D semiconductors, we have already reported on exciton-plasmon conversion effects in our previous paper [Lee *et al.*, Adv.Opt.Mat.3,943,2015], which was also cited in our manuscript [Ref.10]. In this paper, we estimated the exciton-to-plasmon conversion efficiency to ~32% for the AgNW/MoS₂ hybrid using systematic SPP propagation length measurement depending on laser beam position. Moreover, plasmon-to-exciton coupling effects were also studied.

Figure 2. Effect of propagation distance x on the PL spectra of exciton-coupled SPPs in the Ag NW/MoS₂ hybrid device. a) Optical micrographs of the hybrid device with incident laser spots and their PL images. Green arrow: input laser position; white arrow: PL collection position. b) PL spectra collected at the NW-end (white arrow) with various laser input positions. PL spectra (measured at the same excitation and collection positions) from the bare MoS₂ and NW/MoS₂ overlapping position are displayed for comparison. c) SPP propagation lengths derived from the PL spectra as a function of photon energy. Inset: Peak PL intensity as a function of propagation length x along with the exponential decay curve fitting these data points.

<Lee et. al., Adv.Opt.Mat.3,943,2015>

In addition, in short NW-hybrids, we also observed new interesting phenomena: selective amplification effect of the primary exciton in MoS₂ due to cyclic exciton-plasmon interconversions. Notably, many factors including excitonic wavelength, nanowire diameters, nanowire length, determine the exciton-plasmon interactions. For more detail, please see the

reference [Lee et al. Phys. Rev. Lett. 115,226801 (2015)] which was also cited in our manuscript.

FIG. 1 (color online). (a) Schematic of the experimental setup with a side view of the hybrid. PL signals were collected from the NMOR (on NW) and from the bare MoS₂ (off NW) that were excited by an input laser. (b) (top) Optical micrograph showing the LIP (green arrow) and (bottom) PL image showing the collection position (white arrow) of the PL signal at the same LIP. NW length, $\sim 4 \mu\text{m}$. Effective NW length from the NWEF to the LIP, $\sim 3 \mu\text{m}$. (c) Normalized PL signals as a function of P_{ex} for on NW and off NW, with examples of Lorentzian deconvolution at $P_{\text{ex}} = 5 \mu\text{W}$. (d) PL spectra for on NW and off NW at $P_{\text{ex}} = 100 \mu\text{W}$. (e) The log-log scale PL intensity (I_{PL}) as a function of P_{ex} derived from the PL spectra for off NW. (f) The PL spectra for off NW at $P_{\text{ex}} = 5 \text{ nW}$ and 100 nW and SLF for $P_{\text{ex}} = 5 \text{ nW}$ identified as A^0 .

< Lee et al. Phys. Rev. Lett. 115,226801 (2015)>

To emphasize these references, we added the regarding contents in introduction part (page 3):
 “Moreover, the efficient excitons-to-plasmon conversion effects with a conversion efficiency as high as $\sim 32\%$ have been demonstrated in Ag-NW/TMD hybrids^{25,26}.”

2- One fundamental question that I feel needs to be answered is whether the carrier density changes controlled by the gating and its effect on the photoluminescence is the predominant source of changes in the signal or the role it may play in the coupling of the SPP with the TMDs. This can be also understood by better understanding of the yield.

3- I think baseline measurements on TMDs and the Ag-NW is very important and is lacking. I suggest that the authors acquire direct PL from the TMDs and examine the SPP propagation characteristics of the Ag-NW independently and report that in the SI.

4- Similarly, baseline measurements showing the yield of signal conversion in different Ag-NW TMDs seems useful. It is very useful to show if the strength of the excitonic plasmonic interactions are material system dependent.

Author’s answer: To answer the reviewer’s comment 3 and 4, we conducted the additional experiments and the results were added in Supplementary Note 7:

“**Note 7: Material dependence of the exciton-to-plasmon coupling.**

We compared the exciton-to-plasmon coupling effect for three different materials. Figure S7a shows optical micrographs under a laser illumination and their PL images for the Ag-NW overlapped WS₂ (left), WSe₂ (mid), and MoS₂ (right). Under laser illumination at the NW/TMD overlapped regions, PL signals were collected at the input position of laser (Laser-

in, green arrow) and the position of NW end (SPP-out, white arrow) for each sample. For comparison, PL spectra were also measured at the bare TMD regions (Bare, orange arrow). The measured spectra are displayed in Fig. S7b. The PL intensity trend ($WS_2 > WSe_2 > MoS_2$) and peak shape are similar for the three different positions of the sample. Figure S7c shows comparative maximum PL intensity curves obtained from the PL spectra of Fig. S7b. The PL intensity curves for the Bare and Laser-in are in the similar values. While the PL intensity values for SPP-out are prominently lower than those for the Bare and Laser-in due to the coupling and SPP propagation losses, all curves show similar intensity trend of $WS_2 > WSe_2 > MoS_2$.

For each sample, the SPP losses are estimated to be the similar order of magnitude ($\sim 55\%$ to $\sim 65\%$) via FDTD mode analyses: the SPP propagating length (from laser input to NW end) for each sample ranging from $\sim 8 \mu\text{m}$ to $\sim 12 \mu\text{m}$, the SPP losses for each excitonic wavelength ranging from $600 \text{ nm} \sim 750 \text{ nm}$. Because of this we assumed that the deviation of SPP propagation losses for each sample is negligible. We conclude that the material dependence of exciton-to-plasmon coupling effects is negligible although the quantum efficiencies of each TMD are prominently different, since the PL intensity of the SPP-out is quantitatively proportional to that of the Laser-in without material dependence (Fig. S7c).

Figure S7. a, Optical micrographs under a laser illumination and their PL images for three different NW/TMD hybrids. The Ag-NW is partially overlapped on each WS₂ (left), WSe₂ (mid), and MoS₂ (right), respectively. Green arrow: laser input position. White arrow: NW end position. Orange arrow: bare TMD position. **b**, PL spectra collected at the three different

samples. **c**, Comparative maximum PL intensities obtained from the PL spectra (**b**) for each TMD.”

From our experimental results, we could not observe strong material dependence on exciton-plasmon coupling efficiency. However, “*the strength of the excitonic plasmonic interactions*” strongly relies on the exciton quantum yield of each material, as the reviewer suspected. This was mentioned in the revised manuscript at page 5:

“In addition, the 2D material dependence on the exciton-plasmon interconversion efficiency was not appreciable, while the strength of the exciton-plasmon interaction strongly relies on the exciton quantum yield of each material (Supplementary Note 7).”

On the basis of these additional experimental results, exciton Plasmon coupling efficiency does not rely on the quantum yield of the materials. This indicates that coupling of the SPP is not influenced by the gate voltage or photoluminescence intensity. For this, we added or modified some part in the main manuscript:

i) We fixed Fig. 1 explanation at page 3 as “The SPP₀ propagates along the NW in the axial direction with a tightly confined optical near-field^{20,26,29} (Fig. 1b). **The near-field of SPP₀ is absorbed in MoS₂ layers** and excites excitons ($\lambda_1 \approx 660$ nm) at the NW/MoS₂ overlapping region^{26,30} (Supplementary Note 1), where the exciton fluxes are modulated by gate bias (V_G) via excess carrier doping^{12,24}.”

ii) Moreover, we added more sentences for the exciton modulation mechanism regarding Fig.1 at page 3: “The Fermi level was calculated from the excess electron and hole densities derived from the back-gate capacitance as a function of V_G , as plotted in Figure 1e, where the charge neutrality point near -50 V is taken as a reference²⁴ (see Supplementary Note2), **consistent with the intrinsic n-type doping state in MoS₂**. While the Fermi level increases due to the increased electron carriers at high $+V_G$, the PL intensity gradually decreases in proportion to E_F , which is attributed to the Pauli blocking effect for excitons²⁴. **With increasing n-doping ($V_G > -50$ V), the photoexcited electrons are suppressed by Pauli blocking effect and do not contribute to excitonic emission. Consequently, neutral excitons decrease and negative trions increase. Conversely, with increasing p-doping, the Pauli blocking disappears. As a result, both neutral excitons and positive trions increase, and thus, a total exciton flux for p-doping case is larger than that for n-doping case (Supplementary Fig. S3-1).”**

In addition this will be useful in understanding the role that the non-interacting SPP signal may play in the device architecture and signal propagation in the multiplex systems. I also suggest doing experiments that change the order of the TMDs in the signal propagation. This will be useful in answering some of these questions.

Author’s answer: We already demonstrated this experiment suggested by the reviewer in Supporting Information S5:

“The PL intensity of WSe₂ is still higher than that of WS₂ because λ_2 -coupled SPPs are not absorbed in the WS₂ layer. This result implies that electrically modulated signals from each TMD can independently generate and the signals deliver when dielectric spacers between the

TMDs and the NWs electrically isolate each TMD. Here, λ_2 and λ_1 are converted from λ_0 at each TMD positions.”

5- Electrical detection device shown in figure 4C is very difficult to understand. I see three electrodes two of them labeled source and drain. The source seems to be connected to the 2 flakes and Drain is connected to one and the third lead is connected to the second flake. Since Ag/NW is a metal the two flakes are also electronically connected. I am not sure what are the potential levels applied on each lead but I think that these measurements are probably done incorrectly and there is a chance that they are simply measuring a photo detection of λ_0 .

Author’s answer: In the attached figure from the original manuscript, Flake1 and Flake2 are interconnected via Ag-NW. According to the reviewer’s comment, the Ag-NW is metallic so that Flake1 and Flake2 are electrically connected. Thus, when the drain bias (D) is applied, drain current can flow through two paths: Blue arrow for current flow $S \rightarrow \text{Flake1} \rightarrow D$ and Red dashed arrow for current flow $S \rightarrow \text{Flake2} \rightarrow \text{Ag-NW} \rightarrow \text{Flake 1} \rightarrow D$. In this situation, when λ_0 is illuminated on AgNW/Flake2 overlapping position, photocarriers excited by λ_0 in Flake 2 are transported via AgNW and the AgNW acts as an electrode in this case. As a consequence, two types of photocurrents from 1) normal λ_0 absorbed in Falke2 and 2) Flake2-induced-plasmons absorbed in Flake 1 contribute to the detected electrical signals. These are reviewer’s comments.

However, in our preliminary test, the electrical contact between Ag-NW and MoS₂ is poor because the contact area is one-dimensional and they are just physically contacted due to van der Waals interaction. On the basis of this argument, we ignored the current flow $S \rightarrow \text{Flake2} \rightarrow \text{Ag-NW} \rightarrow \text{Flake 1} \rightarrow D$ (red-dashed arrow).

We admit that this is not true in general. To clarify the reviewer’s comment and prevent controversial argument, we simplified the device to demonstrate for the electrical detection of plasmons exclusively, as shown below (new Fig. 4 in the revised manuscript).

<Previous Fig. 4>

<Revised Fig. 4>

Indeed, in previous device structure (previous Fig. 4), the basic concept of the electrical modulation of plasmon flux and the exciton-to-plasmon conversion was overlapped to some degree with other demonstrations (Fig. 2 and 3). Therefore we just focused on the electrical detection itself in the new figure. We revised the Fig. 4 and added regarding content in the revised manuscript. In the simplified architecture (revised Fig. 4), the On/Off ratio for electrical detection of plasmons reaches to ~ 190 at $P_{ex} = 5$ mW and $V_D = 1V$, which is improved significantly compared to the previous geometry. At this point, we really appreciate for the reviewer's comments to improve our manuscript significantly.

At this point I do not recommend the paper for publication but I think with major revisions and addition of new measurements to address some of my concerns this paper can be considered for publication.

Author's answer: Overall, we would like to appreciate for the reviewer's efforts in timely reviewing our manuscript and the valuable remarks. We have made all necessary revisions based on these comments, which made our manuscript stronger.

Response to Reviewer #3

The authors reported the interesting system of a silver nanowire overlapping on the TMDs monolayer field-effect transistor, operating as on-chip optical communication device via exciton-plasmon interconversion. Their experiments seem straightforward and based on their previous results. I wonder if they can provide their answers on my comments and questions below.

Author's answer: We appreciate the reviewer's valuable and constructive comments for our manuscript. The reviewer has already seen our main points and the importance of our manuscript. Thank you very much for your positive comments and concerns. The followings are the details of the responses.

1) The authors used the laser excitation and SPP energy as 514 nm, which is quite far from the MoS₂ exciton resonance (660 nm). I wonder if the authors performed the same experiments with the lower laser excitation near MoS₂ exciton, for example, such as a resonant condition between MoS₂ exciton and SPP. If so, I also wonder their results which may help for understanding exciton-SPP interconversion further.

Author's answer: The reviewer suggested to use different light sources “the lower laser excitation near MoS₂ exciton”. If we use the laser wavelength with 660 nm (MoS₂ exciton wavelength), there are three problems in the experimental procedures.

1. Full-width-at-half-maximum (FWHM) of MoS₂ exciton is broad (~50 nm) so that it is difficult to match with that of laser which is very narrow (<~10 nm).
2. The excitonic peak position and FWHM of MoS₂ are strongly modified depending on excitation laser power by generating additional multiexcitons. Please see the reference [Lee et al., Phys. Rev. B., 93, 140409(R) (2016)]. This implies that while we play with laser energy, laser power is another input variable to tune the laser energy, making the problems even worse. This is not necessary to demonstrate the basic concept of exciton-plasmon interconversion, as we did in our experiments.
3. A measurement of pure excitonic emission signal is impossible. In general, during exciton signal measurement we should use band pass filter to block the signal from the input laser source to remove the laser effects such as scattering and reflection. If we excite the MoS₂ excitons using exactly the same laser energy without band pass filter, there is no way to get pure signal from materials. This is not a common practice for PL measurement for this reason. This could be possible with PLE but is out of scope in our demonstration.

For these reasons, the experiment suggested by the reviewer is not relevant to our work. However, to help the reviewer's understanding, we suggested to read our previous works on NW/MoS₂ hybrids. In short NW-hybrids, we observed new interesting phenomena: selective amplification effect of the primary exciton in MoS₂ due to cyclic exciton-plasmon interconversions. Plasmons with the same energy with MoS₂ excitons re-excite the MoS₂ exciton and consequently strong resonant enhancement in emission can be realized. For more

detail, please see the reference [Lee et al. Phys. Rev. Lett. 115,226801(2015)] which is already cited in our manuscript. See below for the reviewer's information.

FIG. 1 (color online). (a) Schematic of the experimental setup with a side view of the hybrid. PL signals were collected from the NMOR (on NW) and from the bare MoS₂ (off NW) that were excited by an input laser. (b) (top) Optical micrograph showing the LIP (green arrow) and (bottom) PL image showing the collection position (white arrow) of the PL signal at the same LIP. NW length, ~4 μm. Effective NW length from the NWEF to the LIP, ~3 μm. (c) Normalized PL signals as a function of P_{ex} for on NW and off NW, with examples of Lorentzian deconvolution at $P_{ex} = 5 \mu W$. (d) PL spectra for on NW and off NW at $P_{ex} = 100 \mu W$. (e) The log-log scale PL intensity (I_{PL}) as a function of P_{ex} derived from the PL spectra for off NW. (f) The PL spectra for off NW at $P_{ex} = 5 nW$ and $100 nW$ and SLF for $P_{ex} = 5 nW$ identified as A^0 .

< Lee et al. Phys. Rev. Lett. 115,226801(2015)>

2) I understand that the laser located at the end of the silver nanowire can excite the propagating SPP along the interface between nanowire and dielectrics (SiO₂ or TMDs monolayer) via the scattering of the laser light. However, in case of exciton multiplexer and electrical detection experiments, the author focused the laser at the middle of the silver nanowire, in which there seems no way to compensate the momentum mismatch to excite SPP. If the authors want to claim that SPP can be excited via the emission from the TMDs exciton near silver nanowire, I wonder what kinds of mechanism can explain SPP excitation in this case.

Furthermore, it is known that excitons in TMDs are tightly aligned on the plane of the monolayer as described in figure 2a, 3b and 3c, and I guess that the PL from the TMDs excitons will be heading for vertical direction with respect to the monolayer. In this case, the mechanism of SPP excitation is still questionable since it will be similar to the laser excitation shining on the middle of nanowire.

Author's answer: In the previous work on exciton-plasmon interaction using QDs/plasmon hybrids systems [H.Weil et. al, Nano Lett.9,4168(2009); M. Miyata et al. Opt. Express 21,7882(2013)], it is known that the excitons can be coupled to SPP without momentum matching. Please see the following captured sentence and image.

cal functionalization. When the QDs are optically excited, emission from the QDs directly couples to long guiding SPPs supported by the thin metal film, without specific polarization and momentum matching; it results in the emission from an output slit (Fig. 1). The excited SPPs

Fig. 1. Direct coupling between QDs and SPPs on a thin metal film. (a) Schematic of the QD-based plasmon emitter. SPPs can be excited directly by illuminated QDs, resulting in the emission from an output slit. (b) Energy structure of the QD-SPP coupling system. The arrows show the transitions. Photons are first converted into excitons of QDs, and a part of the excitons can be coupled to SPPs in thin metal films. The excited plasmons then propagate and finally are absorbed in metal layers or coupled to far fields at an output coupler.

<M. Miyata et al. Opt. Express 21(2013)7882>

In fact, this exciton-plasmon coupling without momentum matching was explained by Förster resonance energy transfer (FRET) which is energy transfer of excitons in semiconductors to plasmonic hot carriers in metals. Please see the reference [j.Li et al. Nat. Photonics 9,601 (2015);C. Jia et al., Adv. Energy Mater. 6,1600431(2016)], and captured sentence and figures.

Figure 6. Resonance energy transfer at the plasmonic metal/semiconductor interface. For plasmon-induced resonance energy transfer, the plasmon is excited and its energy is transferred to the semiconductor (left). For Förster resonance energy transfer (FRET), the semiconductor is excited and its energy is transferred to the plasmon (right). Reproduced with permission.^[41] Copyright 2015, Nature Publishing Group.

<C. Jia et al., Adv. Energy Mater. 6,1600431(2016)>

We added the regarding sentence in the revised manuscript at page 4:

“The λ_1 is coupled to SPP1 near the NW (Fig. 3b) without momentum matching via Förster resonance energy transfer of excitons to plasmons^{21,32}”

3) The authors explained that the charge neutrality point is -50 V. If this is the case in their experiments, I wonder how to explain the increased PL for the gate bias smaller than -50 V (from -100 V to -50 V) since it is expected the excitonic transition strength being reduced in this condition.

Author’s answer: The reviewer’s question is why PL intensity increases with p-doping (V_G

from -100 V to -50 V). With increasing n-doping ($V_G > -50$ V), the photoexcited electrons are suppressed by Pauli blocking effect and do not contribute to excitonic emission. As a consequence, neutral excitons decrease and negative trions increase. Conversely, with increasing p-doping, the Pauli blocking disappears. As a result, both neutral excitons and positive trions increase, and thus, a total exciton flux for p-doping case is larger than that for n-doping case. Detail on the electrical modulation of MoS₂ exciton was investigated in our previous work for identifying excitons at room temperature [Lee et al., Phys. Rev. B., **93**, 140409(R) (2016)] which was already cited in our manuscript.

To reflect the reviewer's comments, we added the regarding sentence in Fig. 1 (page 3):

“While the Fermi level increases due to the increased electron carriers at high $+V_G$, the PL intensity gradually decreases in proportion to E_F , which is attributed to the Pauli blocking effect for excitons²⁴. With increasing n-doping ($V_G > -50$ V), the photoexcited electrons are suppressed by Pauli blocking effect and do not contribute to excitonic emission. Consequently, neutral excitons decrease and negative trions increase. Conversely, with increasing p-doping ($V_G < -50$ V), the Pauli blocking disappears. As a result, both neutral excitons and positive trions increase, and thus, a total exciton flux for p-doping case is larger than that for n-doping case (Supplementary Fig. S3-1).”

4) If the authors want to emphasize their device mechanism as exciton-plasmon interconversion, polarization (TM, TE modes with respect to the nanowire long axis) dependent experiments are needed to make more clear correlation between exciton and SPP since SPPs in the metal nanowire are known to be sensitive to the TM and TE polarizations.

Author's answer: To answer the reviewer's comment, we conducted additional experiments and added the regarding contents in Supplementary Note 8. This was mentioned in the revised manuscript at page 5 :

“We also investigated the polarization effect of light illumination. A parallel polarization of incident light to a SPP propagation direction along the Ag-NW is essential for the exciton-plasmon interconversions (Supplementary Note 8).”

“Note 8: Polarization effect in exciton-plasmon interconversion.

We investigate the polarization effect of light illumination in the exciton-plasmon interconversion using a partially overlapped Ag-NW on WS₂ flake. Figure S8a and b show optical micrograph and PL images under laser illumination at the NW end for parallel (Fig. S8a) and perpendicular (Fig. S8b) polarizations to the NW direction. The laser light coupled SPP propagates along the NW and excites excitons at the NW/WS₂ overlapping region. While a strong red color emission at the NW/WS₂ overlapping region is observed for the parallel polarization (Fig. S8a, bottom), the emission is negligible for the perpendicular polarization (Fig. S8b, bottom). These results are consistent with the polarization effects in Ag-NWs in previous reports¹².

Figure S8 c and d show optical micrograph and PL images under laser illumination at the NW/WS₂ overlapping region for parallel (Fig. S8c) and perpendicular (Fig. S8d) polarizations to the NW direction. Excited WS₂ excitons at the NW/WS₂ overlapping region propagate along the NW and scattered out at the NW ends. While a strong red color emission at each NW end is observed for the parallel polarization (Fig. S8c, bottom), the emission is negligible for the perpendicular polarization (Fig. S8d, bottom). The results are consistent with Fig. S8 a and b.

Figure S8. The Ag-NW is partially overlapped on WS₂. The polarized laser beams are illuminated at the NW end (a and c) and the NW/WS₂ overlapping region (b and d). Red arrow: parallel (a and c) and perpendicular (b and d) polarizations to the NW direction.”

5) I wonder if the authors have performed any low-temperature experiments other than the reported room temperature results. Two things can be the reasons. At low temperature, exciton dephasing rate is expected to be much lower than the room temperature and the exciton-plasmon interconversion can be enhanced more. Also, by lowering the temperature, the band-gap of TMDs will decrease and the laser exciton and the exciton energy will get closer than the room temperature.

Author’s answer: The low-temperature experiment is out-of-scope in our work, because our motivation of research is to demonstrate merits of two-dimensional semiconductors for constructing reconfigurable device architectures in nanophotonic integrated circuits. In this sense, one of main advantage of TMD is room-temperature operation which is a stark

contrast to quantum well devices that are operated at only low temperature.

The regarding sentence has been already described in introduction:

“Quantum well (QW)-based excitonic transistors, in which exciton flux excited by photons diffuses along the QW channel and switched by the gate modulation of channel potential, are promising devices to realize high-speed interconnection^{7,8}. Nevertheless, short-lived excitons and the finite exciton binding energy critically limit the operation to low temperatures within the limited channel length of typically less than 3 μm ^{7,9}.”

and in the Fig. 2:

“Notably, the demonstrated channel length reaches $\sim 32 \mu\text{m}$ at room temperature, ~ 10 times longer than that of QW-based excitonic transistors ($\sim 3 \mu\text{m}$) at low temperatures^{7,8}.”

Moreover, at low temperature, unknown and defect-related emission peaks are prominent in TMDs. These peaks are not systematically modulated by electrical gating and not fully understood. In this sense, these unknown peaks are not helpful in data analysis of our basic concept and thus the low temperature experiments is irrelevant in our manuscript. Please see the captured figures from references [You, et al., Nature Physics, 11,477(2015); Shang, et al., ACS nano, 9,647(2015)].

<You, et al., Nature Physics 11,477(2015) >

<Shang, et al., ACS nano 9,647(2015) >

6) There seems to be the summary part missing in the paper.

Author's answer: To answer the reviewer's comment, we added the summary part in the end of revised manuscript (page 6):

“In summary, we have demonstrated the crucial optical components for nanophotonic circuits using the reconfigurable exciton-plasmon interconversion and efficient exciton flux modulation of TMDs in various Ag-NW/TMD hybrid architectures at room temperature. The exciton transistor was realized by interconnecting the Ag-NW to TMD-FETs. The laser-coupled-plasmon propagates through the Ag-NW channel and sequentially excites excitons of TMD-FETs where the exciton flux is modulated by electrical gate doping. Wavelength multiplexing devices were realized by interconnecting Ag-NW to laterally arrayed TMD-FETs having different bandgaps via the sequential exciton-plasmon interconversions. The electrical detections of propagating plasmons with a high On/Off ratio were also realized in the Ag-NW hybridized TMD-FET. Our demonstrations of the reconfigurable exciton-plasmon interconversions in various device architectures pave a way to realize various optical components for nanophotonic integrated circuits having advantages of adaptability in wavelength selection and scalability.”

7) I suggest that the On/Off ratio for the electrical detection experiments should be improved to be published in a high impact journal. I guess that it can be achieved in various ways including reducing the length of the nanowires.

Author's answer: To answer the reviewer's comment, we simplified the device to demonstrate for the electrical detection of plasmons exclusively, as shown below (new Fig. 4 in the revised manuscript).

Indeed, in previous device structure (previous Fig. 4), the basic concept of the electrical

modulation of plasmon flux and the exciton-to-plasmon conversion was overlapped to some degree with other demonstrations (Fig. 2 and 3). Therefore we just focused on the electrical detection itself in the new figure. We revised the Fig. 4 and added regarding content in the revised manuscript. In the simplified architecture (revised Fig. 4), the On/Off ratio for electrical detection of plasmons reaches to ~ 190 at $P_{\text{ex}} = 5 \text{ mW}$ and $V_D = 1\text{V}$, which is improved significantly compared to the previous geometry. At this point, we really appreciate for the reviewer's comments to improve our manuscript significantly.

Overall, we would like to appreciate for the reviewer's efforts in timely reviewing our manuscript and the valuable remarks. We have made all necessary revisions based on these comments, which made our manuscript stronger.

REVIEWERS' COMMENTS:

Reviewer #2 (Remarks to the Author):

Authors have revised the paper in a significant way and have addressed many of the questions that reviewers had. I believe the paper is now ready for publication nature communication.

Reviewer #3 (Remarks to the Author):

The authors have addressed my concerns in a satisfactory manner. I can now recommend its publication in Nature Communications.